# Preparation, Characterization and Evaluation of Flavonolignan Silymarin Effervescent Floating Matrix Tablets for Enhanced Oral Bioavailability

**DOI:** 10.3390/molecules28062606

**Published:** 2023-03-13

**Authors:** Sher Ahmad, Jamshaid Ali Khan, Tabassum Naheed Kausar, Mater H. Mahnashi, Ali Alasiri, Abdulsalam A. Alqahtani, Thamer S. Alqahtani, Ismail A. Walbi, Osama M. Alshehri, Osman A. Elnoubi, Fawad Mahmood, Abdul Sadiq

**Affiliations:** 1Department of Pharmacy, University of Peshawar, Peshawar 25120, KP, Pakistan; 2Mumtaz Maternity Hospital, Hashtnagri, Peshawar 25220, KP, Pakistan; 3Department of Pharmaceutical Chemistry, College of Pharmacy, Najran University, Najran 55461, Saudi Arabia; 4Department of Pharmaceutics, College of Pharmacy, Najran University, Najran 55461, Saudi Arabia; 5Department of Clinical Pharmacy, College of Pharmacy, Najran University, Najran 55461, Saudi Arabia; 6College of Applied Medical Sciences, Najran University, Najran 55461, Saudi Arabia; 7Department of Pharmacy, University of Malakand, Chakdara 18000, KP, Pakistan

**Keywords:** flavonolignan, silymarin, oral bioavailability, effervescent, floating matrix tablets

## Abstract

The convenient and highly compliant route for the delivery of active pharmaceutical ingredients is the tablet. A versatile platform of tablets is available for the delivery of therapeutic agents to the gastrointestinal tract. This study aimed to prepare gastro retentive drug delivery floating tablets of silymarin to improve its oral bioavailability and solubility. Hydroxypropyl methylcellulose (HPMCK4M and HPMCK15), Carbopol 934p and sodium bicarbonate were used as a matrix, floating enhancer and gas generating agent, respectively. The prepared tablets were evaluated for physicochemical parameters such as hardness, weight variation, friability, floating properties (floating lag time, total floating time), drug content, stability study, in vitro drug release, in vivo floating behavior and in vivo pharmacokinetics. The drug–polymer interaction was studied by Differential Scanning Calorimetry (DSC) thermal analysis and Fourier transform infrared (FTIR). The floating lag time of the formulation was within the prescribed limit (<2 min). The formulation showed good matrix integrity and retarded the release of drug for >12 h. The dissolution can be described by zero-order kinetics (r^2^ = 0.979), with anomalous diffusion as the release mechanism (*n* = 0.65). An in vivo pharmacokinetic study showed that Cmax and AUC were increased by up to two times in comparison with the conventional dosage form. An in vivo imaging study showed that the tablet was present in the stomach for 12 h. It can be concluded from this study that the combined matrix system containing hydrophobic and hydrophilic polymers min imized the burst release of the drug from the tablet and achieved a drug release by zero-order kinetics, which is practically difficult with only a hydrophilic matrix. An in vivo pharmacokinetic study elaborated that the bioavailability and solubility of silymarin were improved with an increased mean residence time.

## 1. Introduction

Silybum marinum, the milk thistle, is the major source of naturally occurring silymarin. Silymarin is chemically polyphenolic in nature and is mixed with the existing major compound in about 70%. The chemical structure of silymarin flavonolignan is shown in Figure 1. Silybum marinum is composed of three different flavonolignans moieties, including silybin, silydiann and silychristin, with silybin being the most active one. The most typical usage of silymarin is to treat acute and chronic hepatic diseases. Various toxin and active drugs cause hepatitis, cirrhosis and fatty liver alcoholic diseases. It is also used in the treatment of various types of cancer such as prostate cancer, breast cancer and skin cancer [1,2]. The rapid biotransformation in the liver, with a low solubility, is the cause of the poor bioavailability. It is a good option for a gastro-retentive drug delivery system due to its better solubility in acidic media, short half-life and limited bioavailability [2,3].

The low gastric residence time complicates oral control drug delivery systems. Most of the active drugs are absorbed through the stomach small intestine (upper section); rapid gastrointestinal transit can impede complete drug release in the absorption zone and decrease the effectiveness of the prescribed dosage [4]. The dosage form that stays for more time in the stomach is called gastro-retentive drug delivery systems (GRDDS) [5]. In GRDDS, the active drugs release slowly in a prolonged time. Gastro-retentive floating drug delivery systems (GRFDDS) stay in the stomach because of their low density as compared to that of the gastric fluid, and as a result, gastric emptying is not affected [6].

The active contents of the drug are released slowly according to the requirement, as they float in the stomach fluid for a long time. Systems for floating medicine delivery have significant benefits, as they are less prone to gastric emptying, resulting in reduced subject variability in plasma drug levels [5], and are responsible for the delivery of drugs with low absorption values, increased dosing and reduced patient compliance [7]. GRDDS possess good results for acyclovir, ofloxacin and pantoprazole [8,9,10,11].

Designing and testing gastro-retentive floating effervescent silymarin tablets with the intention of enhancing the drug’s efficacy and stability was the aim of the current work [12]. Its absorption is high from the upper GI tract, while it has low absorption because of its lower solubility in the higher pH of the intestine and short half-life. According to earlier results, gastro-retention can improve the bioavailability of silymarin. Our aim was to develop a successful and marketable formulation by eliminating the shortcomings of the published formulations that have already been described.

## 2. Results

### 2.1. Pre-Compression Parameters

The powder mixture was found to have a tapped and bulk density of 0.55 and 0.45 g/mL; the results closely resemble the previously published data [13]. The powder’s Hausner’s ratio, Carr’s index and angle of repose were all assessed. The mixture was discovered to have excellent flow characteristics, with an angle of repose value between 25 and 30 θ. It was discovered, as shown in Table 1, that the values of the Hausner’s ratio and Carr’s index were 1.14 and 12, demonstrating good flow properties [14].

### 2.2. Post-Compression Parameters

The improved formulation’s hardness was within acceptable bounds, indicating adequate mechanical strength [15]. The optimized formulation’s friability value is less than 1%, which shows that it is sufficiently resistant to mechanical shock and abrasion and exhibits good compactness. All the formulations exhibited the content of uniformity within limits; all the results are shown in Table 2 [16].

### 2.3. Buoyancy Studies

The floating lag time of the optimized formulation (SF10) was determined to be 60 s, and the total floating duration was found to be >12 h, while other formulations exhibited a total floating time of barely up to 12 h. Because sodium bicarbonate is included, the optimized formulation exhibits a short floating lag time, as shown in Figure 2 and Table 3. Citric acid was added to the formulation because the stomach’s pH rises when people eat, providing an acidic environment [17].

### 2.4. Swelling Index

The swelling pattern of the improved formulation was excellent, as shown in Figure 3; this is explicable by a decrease in the amount of the acid soluble polymer. It caused the gel layer surrounding the tablet core to become less viscous. Additionally, because Carbopol 934p has a weak solubility in an acidic environment, its presence increased the porosity of the tablet. This disrupts the soluble polymer’s continuous gel structure, allowing more water to penetrate into the swelling tablet [18].

### 2.5. FTIR

The therapeutic efficacy of a drug could be altered by a potential chemical interaction with a polymer. The spectrum of the drug and the selected formulation was analyzed at 400–4000 cm^−1^ to look into the possibility of chemical interaction, as shown in Figure 4.

The spectra of silymarin (Figure 4d) illustrated bands of phenolic OH at 3300, C-H at 2900, C=O at 1635 cm^―1^ and aromatic C=C at 1500 cm^―1^ and 1456 cm^―1^ [19,20,21]. In the FTIR, the Carbopol peak was observed in Figure 4c for -OH at 3300, at 3000 for alkyl group vibrations, at 2372 for -OH vibrations, at 1697 for the COOH stretch and at 1234 for the carbonyl groups [22]. The HPMC K4 spectrum exhibited a broad peak at 3400 cm^−1^ for the OH stretch, at 2850 for C-H vibrations, at 1600 due to the double bond between the carbon atom and the aromatic ring and at 1053 and 947 for the -CO and -CH groups, as shown in Figure 4a [23,24]. HPMC K15 shows a peak at 3400 for OH vibration, at 2900 for C-H cm^―1^ vibration, at 1600 due to the C=C of the aromatic ring and at 1053 and 947 for the -CO and -CH groups (Figure 4b) [23,24,25,26].

Without any discernible spectral shift, the physical admixture and optimized formulation’s FTIR spectra showed all the peaks related to the drug and other ingredients, as shown in Figure 4e,f. This implied that there might not have been any chemical interactions between the formulation’s constituent parts.

### 2.6. Differential Scanning Calorimetry (DSC)

The drug and other excipients may interact, and this has a significant impact. The DSC curve (Figure 5) of the drug showed peaks at 90, 140 and 170 °C, as reported previously [19]. The DSC curve of the polymer Carbopol showed a peak at 240 °C [27]. The DSC curve of HPMC K4 and HPMC K15 showed a peak at 65 °C [28]. The drug peak was effectively preserved in the floating formulation, with only minor changes. The peak shape and enthalpy are said to be influenced by the material quality, particularly in drug-excipient mixtures. Therefore, these slight variations in the drug’s melting endotherm could result from the excipient. As a result, it was determined that every excipient is compatible.

### 2.7. In Vitro Drug Release

By prolonging the retention time of silymarin, the floating drug delivery system is quite helpful. The study was performed six times (n = 6). At 16 h, the formulation showed a drug release of about 95%. Figure 6 shows the in vitro dissolution profile. For longer times, the drug release should be regulated by the polymer combination used in the study. Continuous silymarin release was found in the formulation under analysis.

### 2.8. In Vitro Release Kinetics

The drug dissolution profile was provided by various models, and the data were examined using different models. The diffusion’s exponent was 0.65. To assess how accurately the model was fitted, the correlation coefficients (r^2^) were used. The r^2^ values for the models were calculated and compared. The coefficient of r^2^ demonstrated that the Higuchi model provided the formulation’s best fit. Since “n” was greater than 0.45 and less than 1.0, there was likely anomalous transport. Table 4 displays the release kinetics data.

### 2.9. Ultrasonic Floating Behavior Study

Ultrasound examinations were performed at 0, 1, 2, 3, 4, 5 and 6 h, as shown in Figure 7a–e, respectively. Figure 7 depicts the floating behavior throughout this study. The floating behavior has also been confirmed by radiology. Additionally, the photographs (Figure 7e) demonstrate that the floating tablet’s GRT has successfully been increased by 12 h or possibly longer. An excellent retention of the formulation is indicated by the floating tablet.

### 2.10. In Vivo Pharmacokinetic Studies

In vivo tests for the standard conventional dose form and the improved floating formulation containing 200 mg of silymarin were carried out on albino rabbits. Following oral delivery, the plasma concentration was examined in the blood. Figure 8 shows the conventional dosage form and the formulation’s plasma concentration–time profile curves. The Cmax for the conventional form was 0.6 ± 0.11 µg mL^−1^, and it was 1.1 ± 0.42 µg mL^−1^ for the formulation. The conventional Tmax was observed to be 1.0 h, and that of the formulation was found to be 4.0 h. The area under the curve was 2.4 ± 0.70 and 7.5 ± 1.26 µg·h mL^−1^ for the conventional form and the formulation. The mean residence time ameliorate was from 5.3 ± 0.91 h (conventional) to 8.6 ± 1.15 h, as shown in Table 5. The conventional form and the formulation varied significantly (*p* < 0.05), according to a statistical analysis employing a *t*-test.

### 2.11. Stability Study

The effect of the temperature and RH on the medication content and physical appearance was checked to assess stability. For 12 months, the formulation was subjected to stability investigations in accordance with ICH regulations by storing it at 25 ± 2 °C/60 ± 5% RH, 30 ± 2 °C/65 ± 5% RH & 40 ± 2 °C/75 ± 5% RH. Table 6 lists the outcomes of the examination of the drug content and the physical characteristics at various temperatures. The formulation’s physical appearance remained constant throughout the stability study’s settings. The drug concentration was found to be in the 98.86–100.06% range, 99.21–100.20% range and 99.36–100.04% range at 40, 25 and 30 °C. The results demonstrated the adequate stability.

## 3. Discussion

The influence of the polymer on the drug release and floating behavior was investigated during the fabrication of gastro-retentive effervescent silymarin floating tablets using polymers of various viscosities. The goal of the floating drug delivery system was to maintain the gastric residence time and drug release for a longer period, with increased bioavailability. Sodium bicarbonate reacts with the acidic dissolving media to produce carbon dioxide from the polymer matrix and gives the floating drug delivery system. Preformulation studies were performed, and FTIR spectra displayed all the characteristic bands of both the drug and the excipients, without any significant spectral shift. This suggested that there was no potential interaction between the components [29,30,31]. The drug’s thermal characteristics and its combination with excipients are of particular importance since they may be used to evaluate how various formulation components interact with one another. As a result, it was determined that the drug is compatible with different components to be utilized in the formulation [31,32,33].

The compressibility characteristics of the formulation were satisfactory. The angle of repose ranged between 25 and 30 °C, indicating a good powder flow [34]. A Hausner’s ratio value below 1.25 indicated optimum flowability [35]. Likewise, a Carr’s index value below 15 also indicated the good compressibility of powders. As a result, the direct compression approach was preferred due to its affordability and production convenience [36]. The post-compression tests conducted for the tablets were observed to be within their respective limits. Additionally, the friability of the manufactured tablets was <1%, ensuring the tablets’ durability throughout the packing and handling operations [37]. There were no obvious variations in the formulations’ diameter and thickness. Each tablet’s weight was within the permissible limits, according to the guidelines of USP 38-NF 33 [38]. The assessment of the content homogeneity of each tablet revealed adequate mixing of the powder to produce a homogeneous mixture. The tablet formulation also met the USP criteria for a drug content of 98–102 %, according to a content uniformity test [36]. As a result, it can be said that, overall, the floating matrix tablets’ physical and chemical tests came back within acceptable ranges, according to USP requirements [39].

The examined FDDS utilized sodium bicarbonate as an effervescent agent that, when in contact with the dissolving media, induced an effervescent response. The polymeric matrix entrapped within the resultant gas experiences a slightly reduced density, which will help the tablets to float. This formulation’s floating characteristics—Floating Lag Time and Total Floating Time (FLT and TFT)—were improved by adding more HPMC, and the results are consistent with those of previous research [40,41]. Without a doubt, raising the effervescent agent’s concentration caused the FLT to decrease. The enhanced gas generation caused by sodium bicarbonate’s quick effervescent interaction with the acidic dissolving media may be the cause of this drop in FLT. The system’s overall density reduced as a result of the developed gas being trapped, and there was an instantaneous floating of the tablets [42]. These findings agreed with past research that suggested that larger levels of the effervescent agent led to enhanced gas generation and, subsequently, quicker floating [43]. Due to the fact that the effervescence reaction only happens when the tablet makes its initial contact with the dissolved media, it has been demonstrated that the amount of the effervescent agent only serves to improve the FLT, and the viscosity of the polymer will be the controlling element for the TFT [44].

The kind and quantity of the polymer employed determines how much the tablets will swell. Because of the high viscosity of HPMC K15M and formation of a thicker gel layer around the tablet, the addition of HPMC K4 led to less swelling of the formulation. In addition, when the concentration of HPMC in the produced formulations decreased, the swelling index decreased. It is also important to note that HPMC has superior matrix integrity maintenance, and due to its increased intrinsic water-holding capacity, it is possible that HPMC swells but does not experience severe erosion. The inclusion of Carbopol increased the tablet’s porosity since it had a low solubility in an acidic environment. This will allow water to diffuse through the swelling tablet by breaking up the soluble polymer’s continuous gel structure [40,45]. When HPMC K4M and HPMC K15M were combined with Carbopol Cp-934p, a formulation that had an excellent swelling pattern was created. This can be explained by a decrease in the quantity of the acid-soluble polymer, which led to a less viscous gel layer surrounding the tablet core.

The rate of drug release from the FDDS helps to explain the polymer effect. Through wetting, hydration of the polymer and matrix disintegration, the polymeric matrix of HPMC releases the drug. The soluble medication in the matrix will also be wetted, dissolved and diffused out, while the insoluble components are kept in the polymeric matrix until the polymer breaks down or erodes. Additionally, Wan and coworkers showed that HPMC with increased viscosity produced the development of a thicker gel layer, with the consequence that the drug release was reduced as a result of the denser gel layer. The impact of polymer viscosity on polymeric chain disentanglement may be the most likely explanation for this phenomenon [46]. At the same polymer concentration, a high-viscosity polymer promotes more chain entanglement than a low-viscosity polymer. A higher-viscosity polymer is hydrated as a result, and a thicker and more intricate gel layer is created, which is challenging to dissolve [47]. The link between the molecular weight, polymer viscosity and polymer chain disentanglement is also a factor that affects the drug release, as demonstrated by Korner et al. [48,49]. The release rate of silymarin followed Higuchi’s equation, as the graph of the percentage of drug release vs. the square root time was found to be linear. The values of (diffusion coefficient) n from 0.45 to 0.89 indicate anomalous diffusion to be the mechanism of release. Considering the objectives of the present study, SF10 was elected as the optimized formulation to achieve a conciliation among the floating behavior (short FLT and prolonged TFT) and controlled drug release properties.

Combinations of various viscosity polymers showed improved drug release behavior while maintaining decent floating qualities. The formulation seemed to have a bi-phasic drug release profile, with the first phase being characterized by an early burst release and the second phase being characterized by a regulated, gradual release. A shorter FLT is also a result of the presence of a low-viscosity polymer (HPMC K4M), which causes an initial quick and larger drug release from the pores generated by gel formation. Next, the formation of strong gel owing to the higher-viscosity polymer controls the drug release, further contributing to the longer TFT, i.e., 12 h [43]. This may be the most likely reason for the observed drug release behavior. These findings concur with those of Hiremath et al., who found that combining a low-viscosity polymer (HPMC K4M) with a more viscous polymer (HPMC K15M) increased the rate at which isoniazid was released [50]. However, the type and number of both the polymers determined the drug release rate. When in contact with a dissolving medium, a larger ratio of HPMC K15M resulted in the creation of a viscous gel, which slowed the rate of drug release [51]. Therefore, a higher ratio of HPMC K4M would result in a greater initial burst release and the development of a less viscous gel, which promotes drug release [52]. The optimized formulation was compared to a conventional dosage form for the pharmacokinetic evaluation of different parameters. An increase in the AUC and MRT was observed for the optimized formulation [53,54]. Improved pharmacokinetic parameters were observed.

## 4. Materials and Methods

### 4.1. Materials

HPMC K4, Carbopol 934P, HPMC K15, NaHCO_3_, Mg stearate and citric acid were purchased from Sigma Aldrich, Germany. The silymarin standard was gifted by Nutrabiotics, Peshawar. Methanol and Acetonitrile HPLC grade were purchased from Fisher, UK.

### 4.2. Preparation of Floating Tablets of Silymarin

Silymarin floating tablets were prepared by using different polymers (HPMC and Carbopol 934p) with NaHCO_3_, as shown in Table 7. The ingredients were sieved after weighing. Mixing was performed thoroughly for 15 min. Mg Stearate was then mixed for 2 to 3 min. Compression was performed (12 mm round flat punches) by a single punch machine of Emmay Enterprise with a final weight of 530 mg, as adjusted with lactose. All the parameters such as weighing, mixing and compression were performed at controlled conditions (temperature of less than 25 °C and relative humidity of less than 25%).

### 4.3. Pre-Compression Parameters

#### 4.3.1. Fourier Transform Infrared Spectroscopy

Compatibility testing between silymarin and other ingredients was performed. Physical mixtures (ratio1:1) of the drug and other ingredients were prepared for study. FTIR spectroscopy was used to conduct investigations on drug excipient compatibility. The spectra were recorded from 4000 to 450 cm^−1^, using Thermo Fisher FTIR (Waltham, MA, USA).

#### 4.3.2. Differential Scanning Calorimeter (DSC)

Calorimetric measurements were performed on the drug and other ingredients. DSC measurements were carried out in a nitrogen environment. Heating was increased by ten-degree increments. The dynamic spectra were recorded by DSC (Perkin Elmar, Waltham, MA, USA).

#### 4.3.3. Micromeritics of the Powder Mixture

The powder mixture for tablets was tested to determine the flow behavior, which included the tapped and bulk density, angle of repose, Hausner ratio and Car’s index.

#### 4.3.4. Angle of Repose

This was determined by the fixed height method. Vertically, a funnel was placed on a paper plane kept horizontally. Powder flows through the funnel. The angle of repose was determined by the following formula:$$\theta =tan^{-1}\;\frac{Hieght}{0.5\;base}$$

#### 4.3.5. Bulk and Tapped Densities

Powder was added to a graduated cylinder in weighted amounts, and then the contents were calculated as bulk volumes. The tapped volume was then calculated by tapping the cylinders 100 times against a hard surface. The following equations were used to determine the bulk and tapped densities:$$\mathrm{Pbulk}=\frac{powder\;mass}{powder\;volume}$$
where *pbulk* = bulk density.
$$\mathrm{Ptapped}=\frac{powder\;mass}{tapped\;volume}$$
where *ptapped* = tapped density.

#### 4.3.6. Compressibility Index/Carr’s Index

Flow characteristics can be measured through Carr’s index, which is calculated from bulk and tapped densities. Good flow qualities are indicated by a low percentage, whereas high Carr’s indices represent the poor flow of the powder.
$$\mathrm{Compressibility}\;\mathrm{index}=\frac{ptapped-pbulk}{ptapped}\times 100$$

#### 4.3.7. Hausner Ratio

The blend’s flow behavior is also represented by the Hausner ratio. It was calculated using the tapped-density-to-bulk-density ratio.
$$\mathrm{Hauser}\;\mathrm{ratio}=\frac{ptapped}{pbulk}$$

### 4.4. Post-Compression Parameters

#### 4.4.1. Weight Variation Test

The individual and average weights of 20 tablets which were selected randomly were reported as the mean ± SD.

#### 4.4.2. Content Uniformity

Tablets (10 in number) were selected randomly and were crushed. The powder equivalent to the weight of 50 mg silymarin was dissolved in 100 mL dissolution media. Filtered samples were then diluted and examined at 288 nm. The percentage of drug content was calculated with the help of the UV spectrophotometer (Lambda 21, PerkinElmer, Waltham, MA, USA) by using the equation. A calibration curve was plotted, as shown in Figure 9. The same method has also been used for the in vitro dissolution study.
$$\%\;\mathrm{Assay}=\frac{absorbance\;of\;standard}{absorbance\;of\;sample}*\frac{concentration\;of\;sample}{concentration\;of\;standard}*100$$

#### 4.4.3. Diameter, Thickness and Hardness

The diameter, thickness and hardness were measured for the tablet using the Monsanto hardness tester.

#### 4.4.4. Friability (F)

The initial weights (*w*_0_) of the tablets selected from each batch were noted, and then the rotation was set at twenty-five revolutions per min for 4 min. After 4 min, the tablets were again weighed. Friability was determined by the following equation:$$\mathrm{Friability}=\frac{w_{0}-w}{w_{0}}\times 100$$
*w*_0_ = initial weight, *w* = final weight

#### 4.4.5. In Vitro Dissolution Study

The in vitro dissolution of the optimized formulation was carried out using a USP Type II apparatus at 50 rpm in 0.1 N Hydrochloric acid as the dissolution media. The temperature was maintained at 37 ± 0.5 °C. Samples were collected at pre-determined time intervals (0, 30, 60, 120, 240, 360, 480, 600, 720, 840 and 960 min). Then, these samples were analyzed by a UV spectrophotometer at a *λ* max of 288 nm, as described in Section 4.4.2.

### 4.5. Floating Tablet Specific Evaluation

#### 4.5.1. Buoyancy Properties of the Tablets

The floating time is defined as the time for a tablet between the immersion and the rise to the upper one-third of the medium. The overall time period during which the tablet remained floating above the gastric fluid was the total floating time.

#### 4.5.2. Swelling Behavior of Tablets

The tablet was placed in dissolution media and observed for swelling behavior. Dissolution media pH 1.2 were used (simulated gastric media, 0.1 NHCl, maintained at 37.0 ± 0.5 °C). Tablets were removed after some time and weighed. Tablet swelling was calculated using the following equation:$$\mathrm{SI}=\frac{w_{t}-w_{0}}{w_{0}}\times 100$$
where *w*_0_ = tablet weight before immersion, and *w_t_* = tablet weight at time t.

#### 4.5.3. Kinetic Analysis of the Release of the Drug

The release mechanism from the silymarin gastro-retentive dosage form was determined by different release kinetic models, including zero-order, first-order, Higuchi and Korsmeyer–Peppas [55]. The best-of-fit model was selected based on r^2^ values.

### 4.6. In Vivo Floating Behavior of the Silymarin Tablet

The study was performed on six healthy volunteers who fasted overnight and excluded any gastrointestinal problems before the study. A floating tablet of silymarin was given to the volunteers. Sonographic images were captured at 0, 2, 4, 8 and 12 h. The participant had unrestricted access to water during the investigations. The radiologist eventually evaluated and verified the study results [56].

### 4.7. Study of the Pharmacokinetic Parameters of Silymarin

A pharmacokinetic study was performed on albino rabbits. The rabbits were divided into two groups (n = 6). One group was administered with a conventional formulation (gifted from nutraceutical “Hepavitum” capsule). The other group received a novel formulation of silymarin. All the rabbits used for the study were quarantined. An equivalent dose was administered to rabbits of both groups. Heparin tubes were used for sampling, and plasma was separated and stored at −20 °C. All the procedures were approved by the ethical committee of the University of Peshawar (applicant # 312/EC/F.LIFE/UOP-2020). The plasma was properly thawed and prepared for analysis by HPLC. In this study, a Sykam High-Performance Liquid Chromatographic system (Germany) was employed, with an ultraviolet visible detector. The Welchrom C18 column (250 mm × 4.6 mm, 5 µ) was utilized. Methanol, acetonitrile and TFA (0.0125 %) (10:40:50 *v*/*v*) make up the mobile phase, with a flow rate of 1.0 mL/min.

### 4.8. Pharmacokinetic Parameters

An analysis of different pharmacokinetic parameters, including the area under the curve, the maximum time to reach peak plasma concentration, the half-life and the tmax, was performed by PK solutions software 2.0.

### 4.9. Stability Studies

The prepared formulation of silymarin was kept in a light-resistant container, and this formulation was stored in a stability chamber maintained at 25 ± 2 °C/60 ± 5% RH, 30 ± 2 °C/65 ± 5% RH and 40 ± 2 °C/75 ± 5% RH for 12 months. The samples were then withdrawn for the periodical evaluation of the drug content and physical appearance.

### 4.10. Statistical Analysis

The data were statistically evaluated by Microsoft Excel and Graph Pad Prism.

## 5. Conclusions

The clinical use of silymarin, which is rich in flavonoids, is limited because of its poor dissolution profile and low bioavailability; however, a gastro-retentive drug delivery system can increase its solubility by retaining it in an acidic medium for a prolonged time. We described the reliable and improved tablet formulation of silymarin with continuous releases for more than 12 h, which was also determined in rabbit’s plasma with RP-HPLC. The silymarin gastro-retentive formulation achieved an improved area under the curve, longer MRT, and longer half-life as compared to the conventional formulation in rabbits. An in vivo floating behavior study was performed in humans with the help of ultrasound because of its non-invasive nature, and this showed promising results.

## Figures and Tables

**Figure 1 molecules-28-02606-f001:**
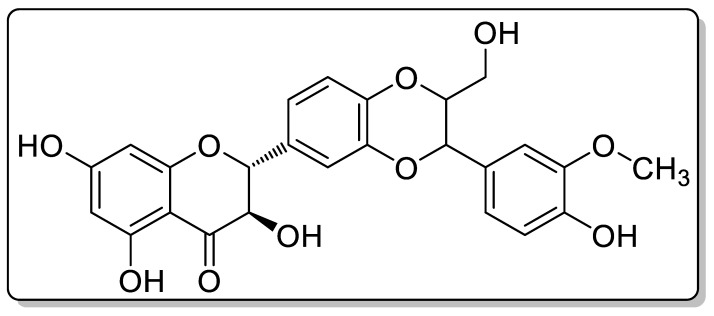
Chemical structure of silymarin flavonolignan.

**Figure 2 molecules-28-02606-f002:**
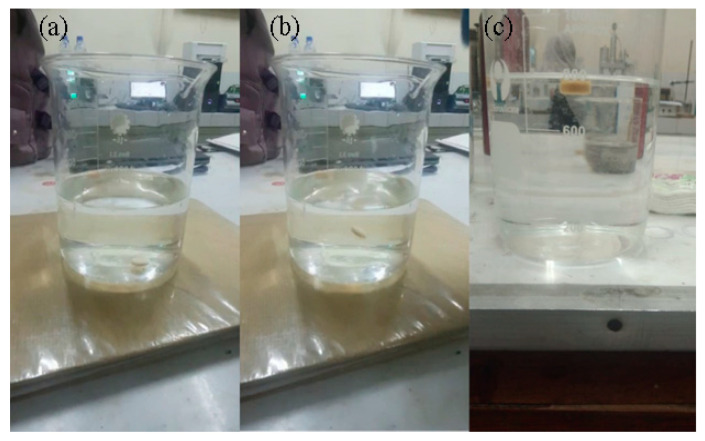
Floating time behavior at times (**a**) 0, (**b**) 55 s and (**c**) 60 s.

**Figure 3 molecules-28-02606-f003:**
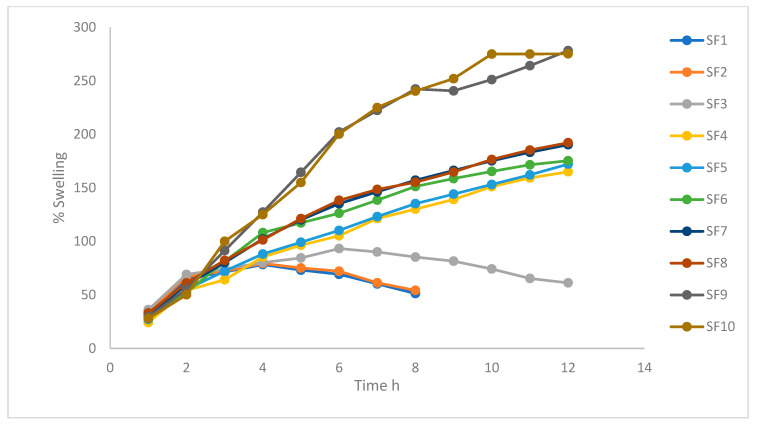
Formulations’ presented swelling index.

**Figure 4 molecules-28-02606-f004:**
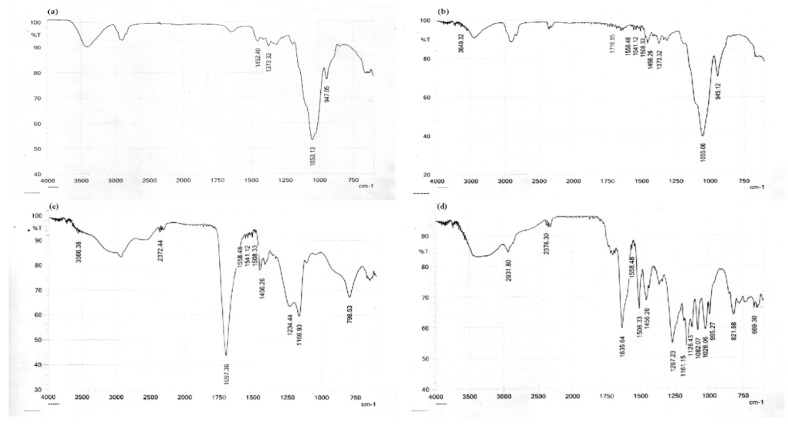
FTIR results of the (**a**) HPMC K4, (**b**) HPMC k15, (**c**) Carbopol 934p, (**d**) drug, (**e**) admixture and (**f**) formulation.

**Figure 5 molecules-28-02606-f005:**
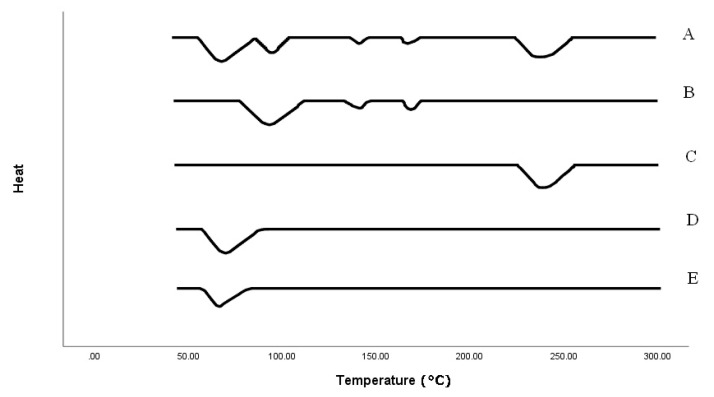
DSC thermograms of the (A) Formulation, (B) Drug, (C) Carbopol, (D) HPMC K4 and (E) HPMC K15.

**Figure 6 molecules-28-02606-f006:**
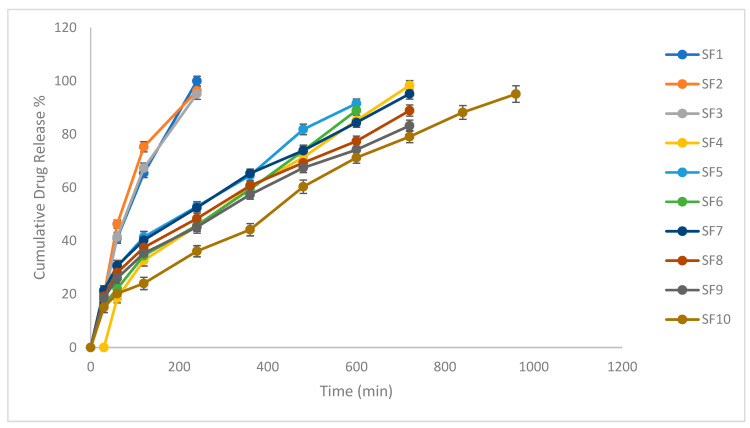
In vitro drug release.

**Figure 7 molecules-28-02606-f007:**
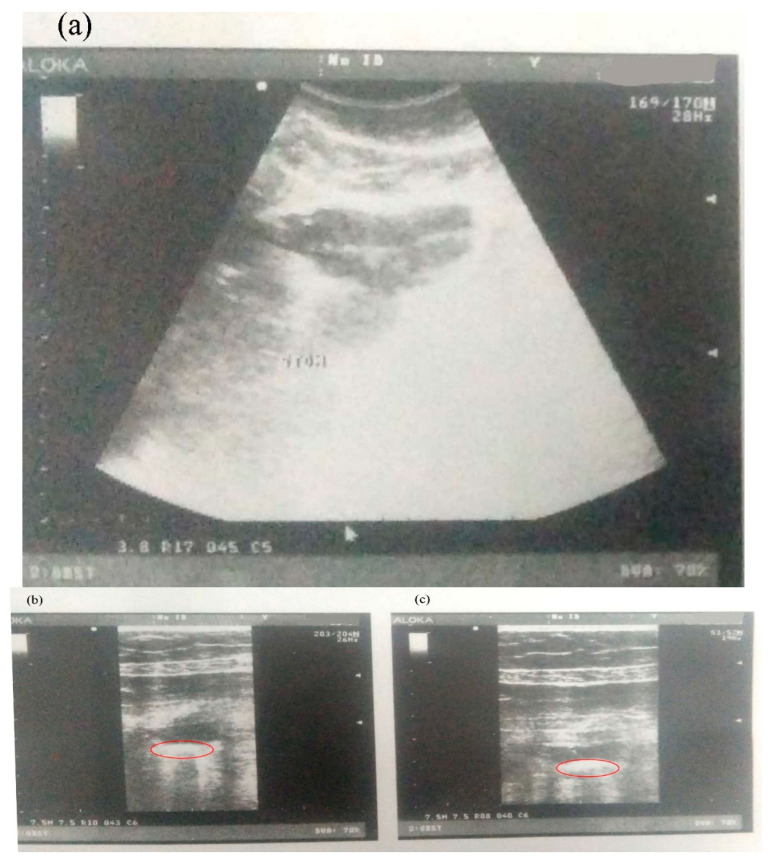
Ultrasound images for floating at (**a**) 0, (**b**) 3, (**c**) 6, (**d**) 9 and (**e**) 12 h.

**Figure 8 molecules-28-02606-f008:**
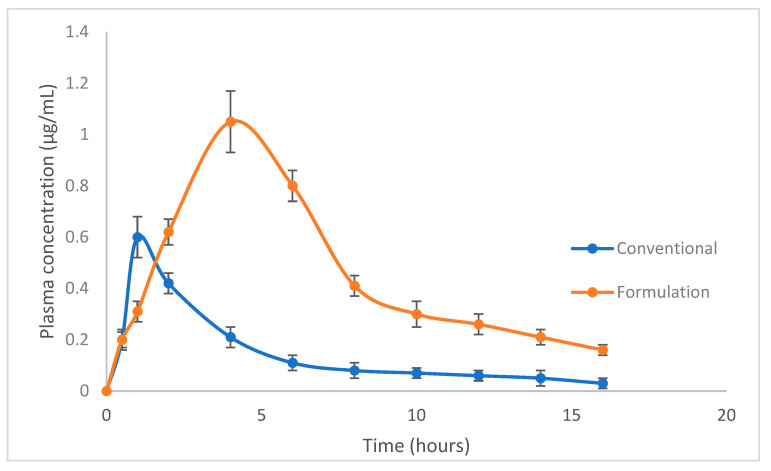
Pharmacokinetic study overlay.

**Figure 9 molecules-28-02606-f009:**
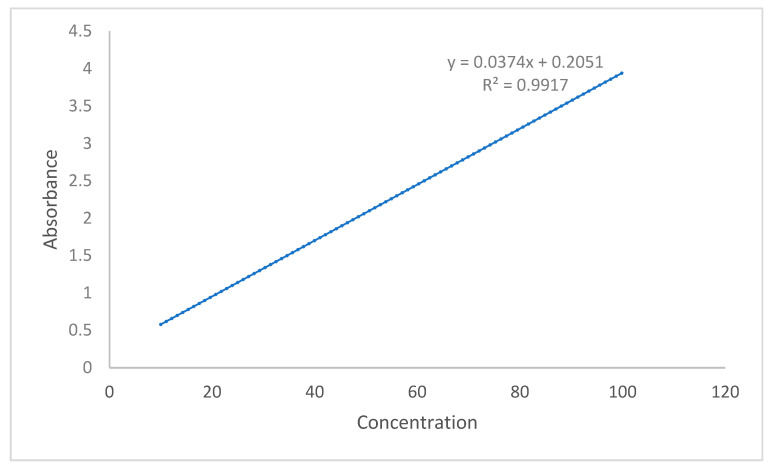
Calibration curve for linearity.

**Table 1 molecules-28-02606-t001:** Pre-compression parameters.

Formulation	Angle of Repose	Bulk Density (g/mL)	Tapped Density (g/mL)	Carr’s Index (%)	Hausner Ratio
**SF1**	25	0.48	0.53	10	1.11
**SF2**	27	0.50	0.52	11	1.12
**SF3**	26	0.47	0.51	09	1.11
**SF4**	26	0.49	0.53	09	1.10
**SF5**	25	0.50	0.59	10	1.10
**SF6**	25	0.50	0.52	10	1.10
**SF7**	24	0.49	0.55	10	1.10
**SF8**	26	0.48	0.58	10	1.11
**SF9**	25	0.48	0.54	11	1.11
**SF10**	26	0.45	0.55	12	1.14

**Table 2 molecules-28-02606-t002:** Post-compression parameters of the tablet.

Formulation	Thickness (mm)	Diameter (mm)	Hardness (N)	Friability (%)	Weight Variation (mg)	Content Uniformity (%)
**SF1**	3.14	12	48	0.32	530	99.1
**SF2**	3.20	12	49	0.34	531	98.8
**SF3**	3.19	12	50	0.33	529	100
**SF4**	3.21	12	47	0.31	530	98.4
**SF5**	3.26	12	49	0.34	532	99.2
**SF6**	3.25	12	45	0.30	531	98.5
**SF7**	3.24	12	46	0.35	530	99.2
**SF8**	3.22	12	44	0.38	528	98.7
**SF9**	3.31	12	45	0.37	527	99.0
**SF10**	3.78	12	52	0.27	530	99.0

**Table 3 molecules-28-02606-t003:** Buoyancy study.

Formulation	Floating Time (Lag) (s)	Floating Time (Total) (h)
**SF1**	34	6
**SF2**	42	6
**SF3**	58	6
**SF4**	70	12
**SF5**	95	12
**SF6**	120	12
**SF7**	50	12
**SF8**	67	12
**SF9**	80	12
**SF10**	60	>12

**Table 4 molecules-28-02606-t004:** Release kinetics results.

Formulation	First-Order	Zero-Order	Higuchi Model	Korsmayer–Peppas	
	r^2^	r^2^	r^2^	r^2^	n
**SF10**	0.587	0.979	0.977	0.986	0.65

**Table 5 molecules-28-02606-t005:** Pharmacokinetic parameters.

Parameters	Dose	Tmax	C_max_	AUC_0-t_	MRT
	mg/kg	h	µg/mL	µg·h/mL	h
Reference	200	1.0 ± 0.08	0.6 ± 0.11	2.4 ± 0.70	5.3 ± 0.91
Formulation	200	4.0 ± 0.12	1.1 ± 0.42	7.5 ± 1.26	8.6 ± 1.15

**Table 6 molecules-28-02606-t006:** Stability study.

Stability Conditions	Sampling Interval (Months)	Physical Appearance	Drug Content (%)
25 ± 2 °C, 60 ± 5% RH	Start	Good	100.2 ± 0.05
	Three	Good	99.86 ± 0.07
	Six	Good	99.52 ± 0.08
	Twelve	Good	99.21 ± 0.12
30 ± 2 °C, 65 ± 5% RH	Start	Good	100.04 ± 0.03
	Three	Good	99.94 ± 0.04
	Six	Good	99.65 ± 0.06
	Twelve	Good	99.36 ± 0.09
40 ± 2 °C, 75 ± 5% RH	Start	Good	100.06 ± 0.07
	Three	Good	99.15 ± 0.14
	Six	Good	98.86 ± 0.15

**Table 7 molecules-28-02606-t007:** Composition of optimized floating formulation.

Ingredients in mg	Formulation Code									
	SF1	SF2	SF3	SF4	SF5	SF6	SF7	SF8	SF9	SF10
**Drug**	200	200	200	200	200	200	200	200	200	200
**HPMCK4**	100	150	200	-	-	-	100	100	100	100
**HPMCK15**	-	-	-	80	100	120	80	100	120	120
**Carbopol 934p**	-	-	-	-	-	-	-	-	-	25
**NaHCO_3_**	70	70	70	70	70	70	70	70	70	70
**Magnesium stearate**	5	5	5	5	5	5	5	5	5	5
**Citric acid**	10	10	10	10	10	10	10	10	10	10

## Data Availability

The data presented in this study are available on request from the corresponding author.

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
