# Peer review of "Preparation, Characterization and Evaluation of Flavonolignan Silymarin Effervescent Floating Matrix Tablets for Enhanced Oral Bioavailability"

_molecules, 2023, doi:10.3390/molecules28062606_

Round 1
Reviewer 1 Report
The author did not describe the room's condition during the preparation of effervescent tablets. Usually, with low humidity.
Reviewer 2 Report
Recommendation: Publish after major revisions noted.
Comments:
Authors prepare gastro retentive drug delivery floating tablets of silymarin to improve its oral bioavailability and solubility. This study is relevant for developing the formulation of drugs with poor dissolution profile and low bioavailability. On the other hand, the experimental results are not well shown. In general, I would recommend publication after a major revision.
Please see my comments below.
1. Line 68, the citation 12 is incorrect.
2. The resolution of Figure 3 and Figure 6 should be improved.The details of these Figures are hard to be obtained in the manuscript.
3. Figure 4, the unit of the x-axis isnot shown.The DSC curves are unnatural, the authors could enlarge the curves or display the original curves.
4. The error bars of Figure 5 and Figure 7 are not shown.
5. The conventional dosage form should be also described.
6. Stability Study section. Impurity should be also considered, not just “Physical Appearance” and “Drug Content”
7. How to obtain the composition of optimized formulation, the authors should add this content.
8. In the section of Pharmacokinetic Parameters of Silymarin, it was performed on albino rabits.while In-vivo floating behavior of silymarin tablet section, why did the authors conduct this study in healthy human firstly, not in animals(rabits)?
Reviewer 3 Report
Silymarin is a natural compound used to treat acute and chronic hepatic diseases. Due to its short half-life, limited bioavailability and better solubility in acidic media, silymarin is a good candidate for a gastro-retentive floating drug delivery system (GRFDDS) and the preparation of such silymarin system was the aim of the current paper. Hydroxypropyl methylcellulose (HPMCK4M and HPMCK15), Carbopol 934p and sodium bicarbonate were used as matrix, floating enhancer and gas generating agent, respectively. A compatibility study between drug and excipients was conducted by FTIR and DSC analysis. The prepared tablets were evaluated for physicochemical parameters (hardness, weight variation, friability, floating properties, drug content, and stability study), in vitro drug release, in vivo floating behavior and in vivo pharmacokinetics. It can be concluded that the new developed silymarin system, containing hydrophobic and hydrophilic polymer, has minimize the burst release of drug from the tablet and achieved a drug release by zero-order kinetics, and improve the bioavailability and solubility of silymarin with the increase of the mean residence time.
The study is well designed and based on the obtained results, the authors can develop further formulation studies on this formulation. The description of the analytical methodology must be improved. However, the results seem to be correct interpreted. Clarifications or additional information are needed, as follows:
· For the tablets preparation, the compression machine type and the compression process parameters (e.g. time, force) should be mentioned.
· What analytical method was used for drug content determination? The method should be described, including calculation formulas.
· The UV spectrophotometric procedure used for content uniformity determination and dissolution study should be described. The quantification method should be mentioned. If external standard quantification method was used, the standard used should be mentioned in materials section.
· The dissolution study description should be improved. The predetermined time intervals should be defined. In order to sustain the study conclusions, the in vitro dissolution results should be presented, not only the correlation coefficients.
· For the swelling behavior of the tablets, it must be clarified and if water is used or dissolution media. The media choice should be justified.
· The RP-HPLC method used in pharmacokinetic study should be briefly described and materials section should be updated if needed.
· Stability studies: the conclusion of adequate stability cannot be fully endorsed if only appearance and drug content were analyzed. As the dissolution test is regarded as the performance test for solid unit-dose preparation, at least dissolution results should be added.
· It is unclear how reference index were cited for the aim or for the actual results of the study. E.g.: [12] on 69 row, [15] on 191 row, [18] on 203 row, [21] on 250 row, etc.
· It is unclear what “improved formulation” is meaning, as only one formulation was presented.
· It is unclear how the informed consent statement was considered “not applicable” as a research that involves human subjects was described.
· More typos were noticed and should be corrected. E.g.: additional spaces on row 47 and 64, “was” instead of “were” on row 75 and 77, a missing “by” on row 76, NaHCO3 on row 83 and in Table 1, Celsius degree everywhere, “thermos fisher FTIR” instead of “Thermo Fisher FTIR”, etc.
· The bibliography must be revised and updated as per journal requirements. 10-th bibliographic item shod be corrected.
Overall, the presentation of the data does not value the multitude of studies carried out and I consider that it must and it can be improved.
Round 2
Reviewer 2 Report
No comments
Author Response
The reviewer 2 has no comment to answer
Reviewer 3 Report
The manuscript was improved, but further corrections/modifications are needed as follows:
Page 3:
“emmay enterprise” should be replace by “Emmay Enterprise”
Table 1 : NaHCO3 should be replace by NaHCO3 and the dot after Mg (Mg.) should be deleted.
“Thermos Fisher FTIR” should be replace by “Thermo Fisher FTIR” (without “s” after Thermo!!!)
Page 4:
The Calibration curve should be Figure 2 and it should be presented in section 2.4.2.
???????? should be replace by ????????
According to the equation, a standard is used. Please present also the Silymarin standard in section 2.1 Materials.
Page 5:
It is still unclear if the swelling behavior of the tablets was performed in water or in dissolution media. If water was used, the dissolution media should be removed from section 2.5.2. If dissolution media was used, than the tablet was not place in water! Please correct accordingly to actual experimental conditions.
The chromatographic column should be further characterized by length, internal diameter and particle size of the C18 stationary phase.
Section 3. Results and Discussion should be rename “Results” as Section 4. “Discussions” was added.
Page 8:
“C=C1500 cm-1” should be replace by “C=C at 1500 cm-1“
“at1697” – a space should be added after at
Page 13
A Discussions section has been added. However, this section refers to some formulation changes were not described in the current paper. Please add the mentioned formulation studies (it seems than at least 10 formulations were studied, and SF10 was elected as the optimized formulation) in Material and Method section and in Results section.
Informed Consent Statement:
“Not applicable” is suitable only for studies not involving humans. Any research article describing a study involving humans should contain this statement. “Informed consent was obtained from all subjects involved in the study.” Please add “Informed consent was obtained from all subjects involved in the study.”
